# Preventive Maintenance Topic Models for LNG Containment Systems of LNG Marine Carriers Using Dock Specifications

**Ik-Hyun Youn [1] and Sung-Cheol Kim [2],***

[1]    Division of Navigation & Information Systems, Mokpo National Maritime University, Mokpo 58628, Korea; iyoun@mmu.ac.kr

[2]    Graduate School, Mokpo National Maritime University, Mokpo 58628, Korea

*    Correspondence: sckim@mmu.ac.kr; Tel.: +82-61-240-7421



**Featured Application: The proposed mathematical maintenance models may be directly applied by the maintenance managers of liquefied natural gas carrier (LNGC) fleets to establish a more reliable and safer LNGC cargo containment maintenance strategy. The proposed framework could also lead to the discovery of higher quality critical system maintenance topic models in various fields such as the construction and manufacturing industries.**

**Abstract:** The high demand for liquefied natural gas (LNG) requires more LNG carriers (LNGCs) to be in operation. During transportation, there is a high risk due to the required extremely low temperatures and the explosive nature of LNG cargo. Moreover, when there is a lack of experience in operating old LNGCs, there is a serious concern regarding operational accidents. A systematic maintenance strategy, especially for LNG cargo containment systems, is crucial for maintaining safe LNG transportation at sea. The purpose of this study is to develop preventive LNG cargo containment system maintenance models by using LNGC dock specifications from LNGCs of various ages. The dock specifications from a conventional LNGC repairing dock were analyzed using natural language processing techniques in order to develop preventive maintenance models of the LNG cargo containment system. From these results, and by considering the ship's age, it was found that for young LNGCs, the priority for repair should focus on checking routine consumable spare parts by tank inspections, whereas for older LNGCs, the focus should be on tank condition maintenance rather than on other facilities. These results are expected to be useful in the development of a maintenance strategy of preventive LNG cargo containment systems in maritime LNG transportation.

**Keywords:** preventive maintenance model; LNG cargo containment system; aging effect; dock specification; natural language processing

## 1. Introduction

Liquefied natural gas carriers (LNGCs) are sophisticated ships designed for the transportation of liquefied natural gas (LNG) in the challenging marine environment. Commercial seaborne LNG transportation has been used for more than 50 years (i.e., since 1964) [1,2]. In recent decades, a dramatic increase in the demand for LNG and the rapid growth in LNG production have led to a large number of new building contracts for large LNGCs to cope with the growing global LNG market. Unlike other marine dry cargo carriers such as container vessels or bulk carriers, LNGCs must consider a reliable LNG cargo containment system for safe LNG marine transportation [3,4]. LNG is kept at extremely low temperatures, and the vapor from LNG is an explosive hazard; hence, cryogenic damage and the risk of catastrophic fire must be reliably managed during the entire lifespan of LNGCs.

Currently, most LNGC maintenance types are scheduled and corrective maintenance to maintain operational status of LNGCs as illustrated in Figure 1. There is a lack of an objective maintenance database for determining appropriate maintenance timing, particularly for LNG cargo containment systems. Dock Specification was one of the available maintenance data including significant maintenance works during a dry docking repair. All maritime vessels are require to perform an inspection of the hull in a dry dock twice within a 5 year period. The dry dock repair includes maintenance of hull, propeller, rudder, cargo containment systems, and other immersed parts during operation. Additionally, the cost of each maintenance was unobtainable due to the confidentiality of companies, so cost-benefit analysis could not be performed.

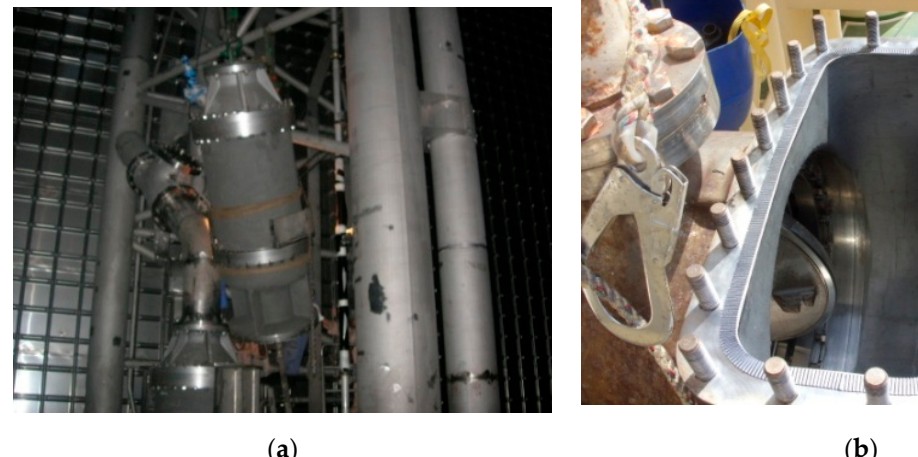

(**a**) (**b**)

**Figure 1.** Examples of liquefied natural gas (LNG) cargo containment system maintenance tasks during dry docking. (**a**) Detaching a cargo pump in an LNG cargo tank for an overhaul inspection. The inside of the LNG cargo tank should be prepared as a nonhazardous working environment after inerting, warming, and aerating. (**b**) Butterfly LNG cargo valve inspection after inerting, warming, and aerating for safe repair.

LNGCs tend to be used for 20-year projects, and LNG cargo containment systems and materials are usually confirmed to operate safely for the initial contract period of the project [5,6]. Because LNG cargo containment systems are made of high-quality materials, aging LNGCs, which are over 20 years of age, are still in practical and commercial use. Many LNGC owners consider extending the practical operation lifespan of LNGCs beyond 20 years. A systematic preventive maintenance strategy during the operational phase is the most recommended method to maintain safe operation and to monitor any defects caused by aging [7–9]. The preventive maintenance of the cargo containment systems of LNGCs is a maintenance strategy that is carried out on equipment before the failure of any critical LNG cargo handling facility. Originally, preventive maintenance was introduced in repairable systems to enhance system reliability [10,11]. Preventive maintenance of LNG cargo containment systems is recommended because the maintenance technique increases the life cycle of the cargo containment system and reduces the risk of cargo containment system failure [12–14]. However, in the case of aging LNGCs, a lack of monitoring data to maintain the LNG cargo containment system contributes to an increase in the risk of safe and efficient maintenance tasks. The fundamental strategy of preventive maintenance of LNG cargo containment systems is to repair any possible defective items based on cumulated condition monitoring data; however, the monitoring data from aging LNGCs are limited. On the basis of the World LNG Report by the International Gas Union, 478 LNGCs were in active operation at the end of 2016; among these, only 50 were more than 20 years old [15]. Therefore, it is difficult to establish an objective and systematic LNG cargo containment system maintenance model owing to the lack of data of aging LNGCs.

The purpose of this study was to develop preventive maintenance topic models for LNG cargo containment systems of LNG marine carriers by using the dock specifications of LNGCs. In the present study, preventive maintenance topic models for LNG cargo containment systems were developed by a natural language processing (NLP) method based on LNGC dock specifications. The order of priority for LNGC repair items was identified from the dock specification documents. The priority trends were categorized into three LNGC age groups from the recognized priority of maintenance topics based on the age of the LNGCs. Finally, preventive LNG cargo containment system maintenance models were developed using the latent Dirichlet allocation (LDA) method to discover the underlying maintenance topics.

## 2. Development of LNGC Maintenance Topic Models

This section illustrates a framework for the development of the proposed LNG containment systems' maintenance topic models. The cargo containment system of LNGCs is not used during operation due to exitance of LNG and its vapor in the containment system. A regular dry docking period is the best opportunity for maintenance of the LNG containment systems after inerting, warming up, and aerating of the containment system have been completed at a service shipyard [16,17]. LNGC dock specifications are documents that contain lists of severe and important ship maintenance tasks, including LNG containment maintenance orders during dry docking. The dock specifications of multiple LNGCs were collected from shipyards in the Republic of Korea, Singapore, and Japan. Table 1 demonstrates the detailed information of collected dock specifications from multiple shipyards. The priority of the LNG cargo containment systems' maintenance topics was identified by using various texture data analytical techniques. Finally, the maintenance topic models were classified into particular topics by the texture data collected from the documents.

**Table 1.** Information collected on dock specifications based on characteristics of liquefied natural gas carriers (LNGCs).

| Characteristics | Types | Number of Dock Specifications | Total |
|---|---|---|---|
| Containment systems | Membrane | 37 | |
| | Spherical IMO type B Tank (MOSS type) | 11 | |
| Propulsion systems | Steam turbines | 46 | |
| | Dual-Fuel Diesel Electric | 2 | 48 |
| Cargo capacity | <150,000 m$^3$ | 43 | |
| | 150,000–180,000 m$^3$ | 5 | |
| Vessel age | >20 | 12 | |
| | 10–20 | 16 | |
| | <10 | 21 | |

Maintenance topic models using the LDA method were developed and analyzed using the collected dock specifications. The key maintenance topics of LNG cargo containment systems were then illustrated as a word cloud and by word-embedding diagrams. Various text analytics techniques were applied to extract valuable insight from the cumbersome documents. In this section, a framework of preprocessing and analyzing the dock specifications for developing LNGC cargo containment maintenance topic models is presented.

### 2.1. LNGC Maintenance Data Analysis

NLP was applied to analyze the collected LNGC maintenance data. NLP is a widely used machine learning technique that includes various types of text extraction, data analysis, and visualization tasks [18,19]. The dock specification for a dry dock repair is not created for further data analysis, so the maintenance data written in the dock specification are unstructured. Natural Language Processing (NLP) is a subfield of artificial intelligence that provides the ability to read and understand the

unstructured text data from the dock specification. This section explains the methods that were used to (i) import textual data, (ii) tokenize texture contents, and (iii) discover insights for developing the maintenance topic models. The series of data analysis processes is demonstrated in Figure 2.

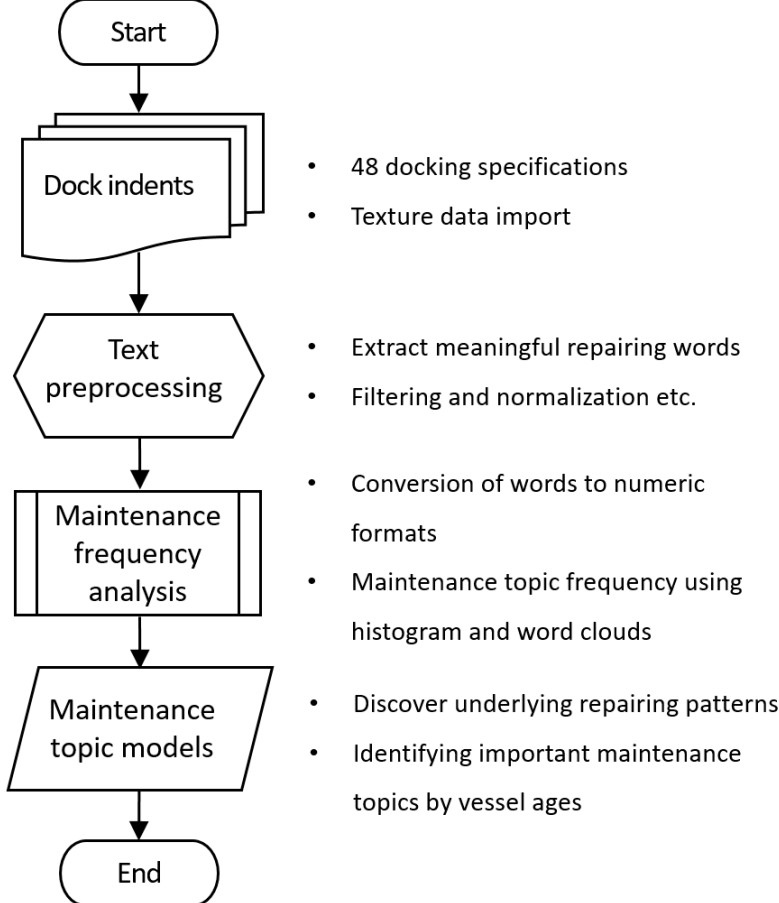

**Figure 2.** Framework for the development of LNG containment system maintenance topic models.

As an initial step to analyze data, the raw data collected from the LNGC dock specification were extracted from different file formats. Various forms of words were then stemmed and indexed using predefined grammar rules [20–22], and the indexed word data set was used in the next data analysis process, including analysis, extraction, and clustering methods. The practical implementation of these processes consists of three phases: tokenization, normalization, and numerical conversion.

First, tokenization is a process that splits the text into sentences and the sentences into individual words [23]. Tokenization primarily splits lengthy strings of text into smaller tokens. The original string of text is initially tokenized into sentences; then, the sentences are tokenized into individual words. Similarly, segmentation could also be used to break down the lengthy text into segments larger than individual words, such as sentences or paragraphs, whereas tokenization results exclusively in individual words.

Second, normalization of the tokenized words was carried out [24]. Although each word is recognized and stored in digital format, various word formats hinder the formation of coherent insights of the texture data. Converting the given string into lowercase, removing punctuation, and removing fewer character words are conducted concurrently during tokenization. Normalization trims all the different word formats on an equal footing; hence, the same expressions are uniformly handled in the data analysis step. Typically, normalization of a text can mean two distinctive tasks: stemming and lemmatization. Stemming eliminates affixes such as suffixes and prefixes from the tokenized word to achieve the word stem, whereas lemmatization extracts canonical forms based on

the predefined lemma of the stem word. For example, stemming the word "electronics" would return "electron", whereas lemmatization of the same word would result in "electronics." This study applied lemmatization to extract the canonical forms of words.

Finally, the tokenized and normalized words were converted into numeric formats for use in the data analysis phase [25]. The calculation of single-word frequency statistics to represent the text data numerically was performed to identify the most frequently occurring maintenance items in the dock specification. The text preprocessing framework explained in this section is illustrated in Figure 3.

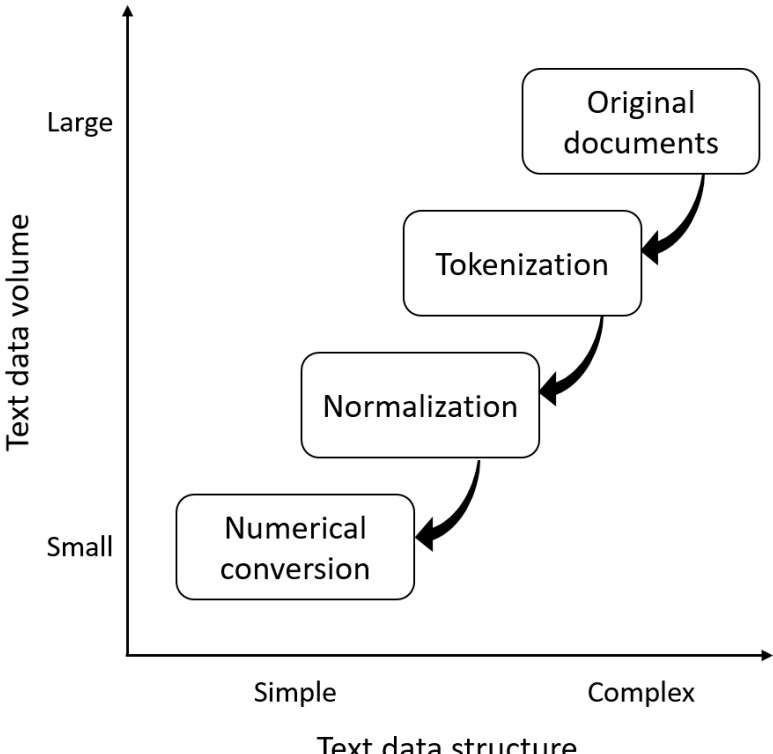

**Figure 3.** Framework for preprocessing dock specification texture data from the collected documents.

The preprocessed data were analyzed to identify latent insights of the maintenance topics. Single-word frequencies were analyzed to examine the distribution of the maintenance items [26]. Histograms were used to represent the probabilistic priority of each maintenance item. Moreover, contiguous sequences of multiple words were extracted from the preprocessed text data. Sequential information is the foundation of data analysis for the co-occurrence and proximity of each maintenance item in the dock specifications.

*2.2. Prescriptive Maintenance Topic Models of LNGCs*

Prescriptive maintenance is an alternative maintenance strategy that fills the gaps in a strictly planned maintenance schedule. Prescriptive maintenance could take advantage of advanced data analytics by identifying maintenance patterns and timings to indicate appropriate and timely repairing actions before unexpected failures. Topic modeling was applied to improve the ability of prescriptive maintenance, particularly for the LNG cargo containment systems.

Topic models provide us with a meaningful collection of topics that make sense as a group [27]. Latent maintenance patterns could be discovered by developing topic models. Topic models were also introduced as probabilistic topic models, which indicates statistical methods for uncovering latent semantic relationships. Topic modeling began with a linear algebra approach [28] known as latent semantic analysis. Later, a foundational probabilistic topic model, the LDA model, was introduced [29,30], which was used in this study. In the LDA model, each dock specification is

considered as a compound of various topics. None of the semantic knowledge extracting techniques is applied to topic models. Instead, topic models apply mathematical relations among words from the numerically converted preprocessed texture data. The developed topic models also provide topic annotation. Using the annotation of each topic, structuring maintenance topics is also available. Several mathematical techniques are used to develop topic models.

## 3. Results

LNG containment maintenance topic modeling was used to explore latent critical system maintenance patterns. Discovering the priority of maintenance keywords based on vessel age categories was performed using histograms and word clouds [31]; developing maintenance topic models based on vessel age categories and summarizing the developed maintenance topics by reviewing the original dock specifications were sequentially performed. To discover the maintenance patterns of aging LNGCs, the results of every data analysis step were compared based on three vessel age categories: younger than 10, between 10 and 20, and more than 20 years of age. The results of the analysis sequence are detailed below.

### 3.1. Frequency Analysis

Single-word frequencies were analyzed to examine the distribution of maintenance items. Histograms were used to represent the probabilistic priority of each maintenance item. Table 2 summarizes the most frequently appearing maintenance keywords in the dock specifications. Three key maintenance items, "check", "valve", and "tank," were the most frequently appearing words in the dock specifications for LNGCs aged less than 10, between 10 and 20, and more than 20 years, respectively. The lower priority word "check" as a root form of "checking" or "checked" appeared more in the less than 10-year LNGC group, whereas the higher priority word "tank" was evident in the aging LNGC category. The term "renewed" appeared only for vessels aged more than 20 years.

**Table 2.** Top-ranking maintenance items based on single-word frequencies.

| Frequency Rank | Vessel Age | | |
|:---:|:---:|:---:|:---:|
| | **Less than 10** | **Between 10 and 20** | **More than 20** |
| 1 | check | valve | tank |
| 2 | cargo | tank | valve |
| 3 | valve | check | cargo |
| 4 | tank | cargo | check |
| 5 | test | pump | gas |
| 6 | system | gas | pump |
| 7 | spare | test | renewed |

A similar tendency was observed in the histogram statistics, shown in Figure 4. The histograms additionally demonstrate the frequencies of the top-used maintenance keywords by rank. For example, the relative rank of the term "valve" was third, first, and second in the increasing age categories; the same term continuously increased as the vessel age increased. The pattern showed that older LNGCs require more LNG cargo valve-related maintenance tasks. A similar maintenance trend was also seen for the "tank"-related maintenance tasks. Maintenance tasks related to "renewed" started to appear for LNGCs aged more than 10 years, and the frequency of "renewed" tasks significantly increased for older LNGCs (i.e., vessels aged more than 20 years).

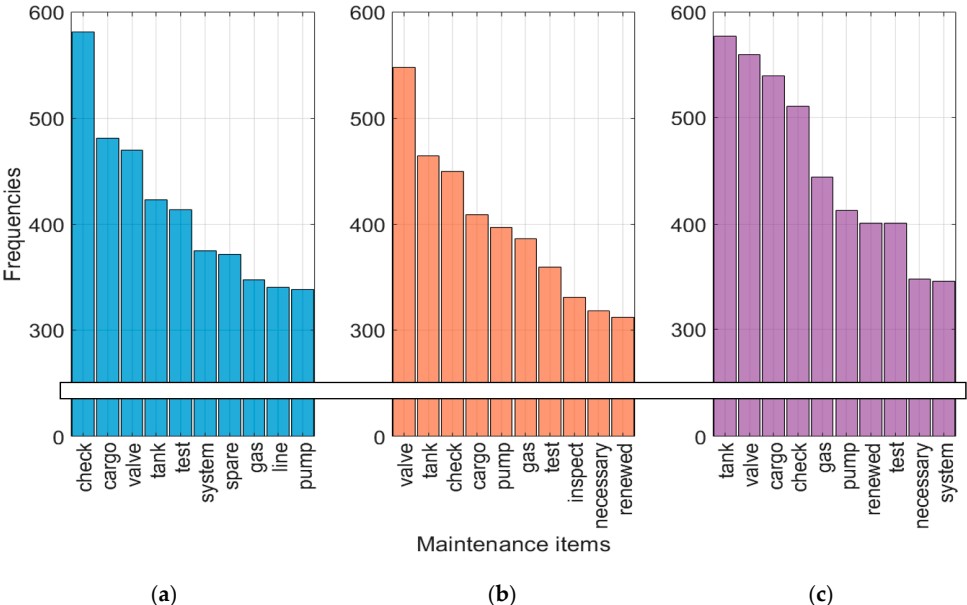

**Figure 4.** Histogram statistics to illustrate relative maintenance item frequencies: vessels aged (**a**) less than 10 years; (**b**) between 10 and 20 years; and (**c**) more than 20 years.

Additionally, the relative significance among maintenance keyword frequencies was analyzed using word clouds (see Figure 5). The maintenance word cloud is an intuitive visual representation of maintenance patterns, mainly used to illustrate key maintenance metadata. The most frequent maintenance keyword is located in the center of each word cloud, and a larger font indicates a higher maintenance frequency. Word clouds are utilized as graphical representations of maintenance keyword frequency in dock specification. The larger the maintenance keyword in the word clouds the more common the word was in the dock specifications.

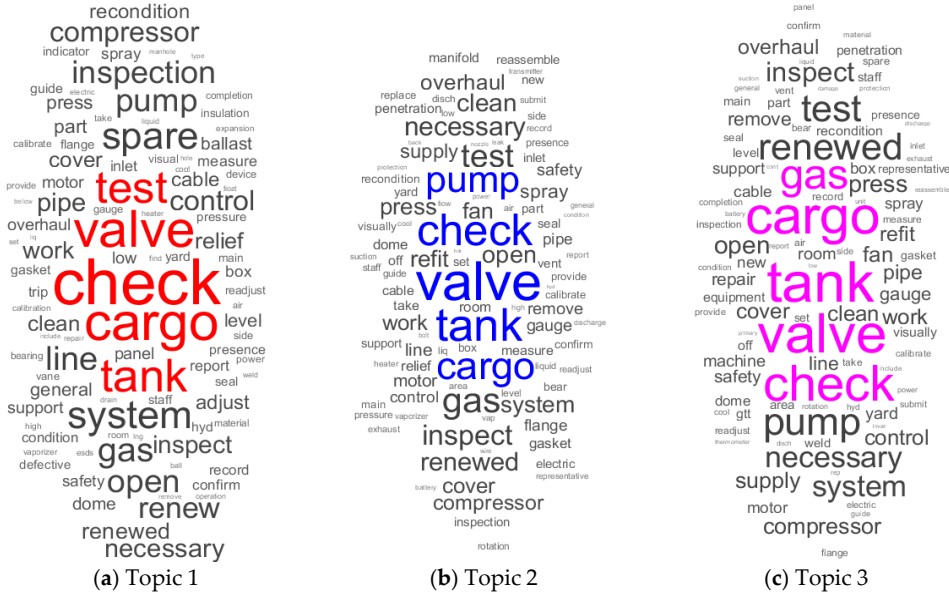

**Figure 5.** Word clouds using single-word frequencies of preprocessing dock specification texture data from the collected documents. Maintenance item word cloud of (**a**) vessels aged less than 10 years, (**b**) vessels aged between 10 and 20 years, and (**c**) vessels aged more than 20 years.

### 3.2. Proposed Preventive Maintenance Topic Models of LNGCs

Topic modeling algorithms were applied to analyze the thematic structure maintenance keywords from the dock specifications. The developed LNG containment topic models were represented using word clouds, and maintenance keywords of higher frequency were highlighted in the LDA topic models. The LDA algorithm determines the key terms with sizes according to probability for the selected LDA topics. Word clouds give a view of the words with the highest probabilities in each topic. Figures 6–8 visualize the first three topics using the word clouds for LNGCs aged less than 10, between 10 and 20, and more than 20 years, respectively.

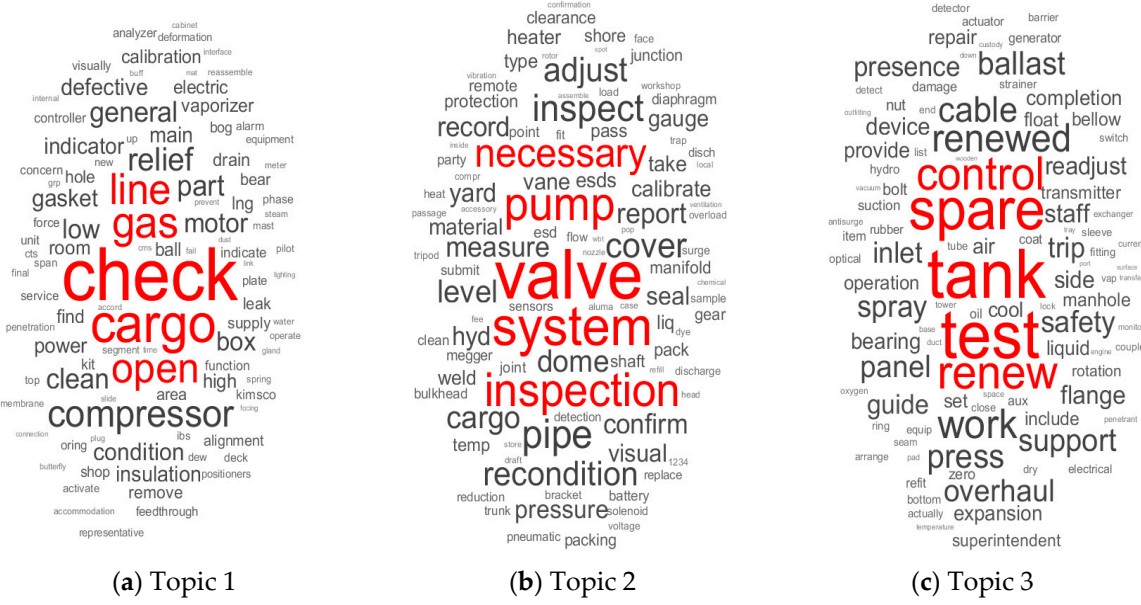

(**a**) Topic 1      (**b**) Topic 2      (**c**) Topic 3

**Figure 6.** Topic model of LNGCs aged less than 10 years. (**a**) Topic 1: check-cargo-gas-line; (**b**) Topic 2: valve-system-inspection-pump; (**c**) Topic 3: tank-test-spare-renew.

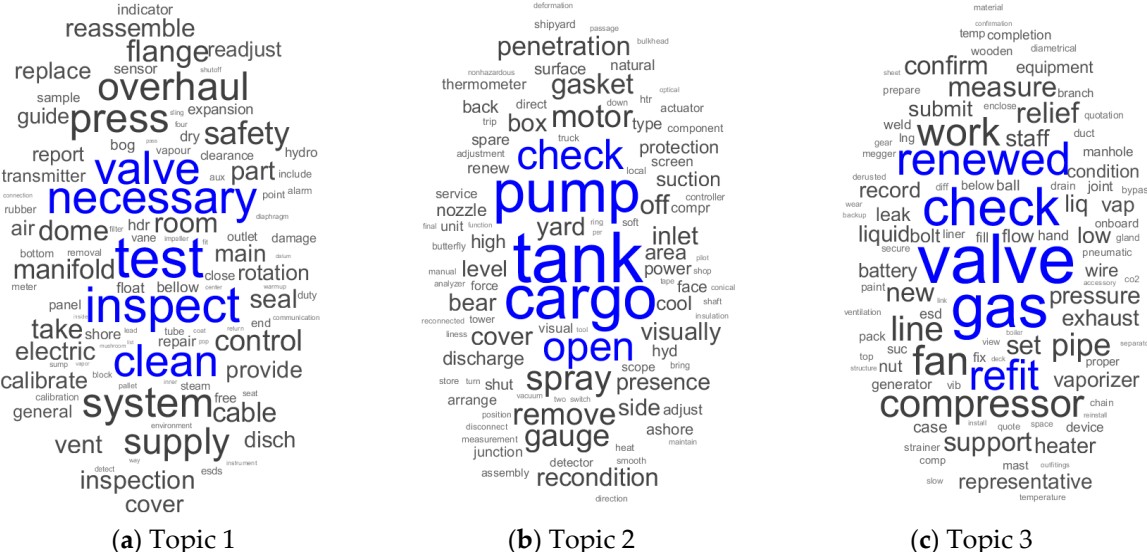

(**a**) Topic 1      (**b**) Topic 2      (**c**) Topic 3

**Figure 7.** Topic model of LNGCs aged between 10 and 20 years. (**a**) Topic 1: test-inspect-valve; (**b**) Topic 2: tank-cargo-pump; (**c**) Topic 3: valve-gas-renew.

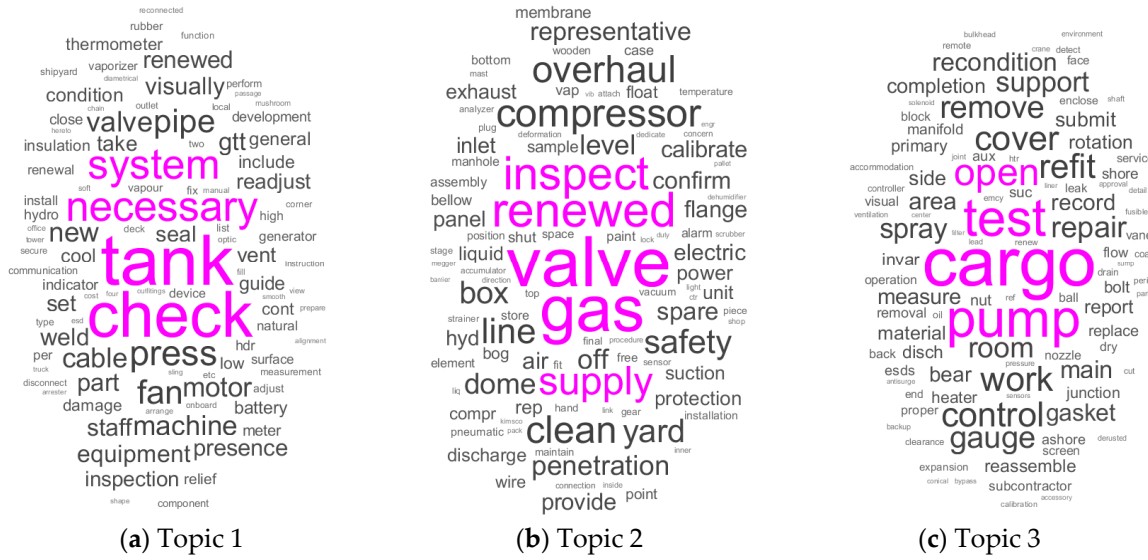

(**a**) Topic 1       (**b**) Topic 2       (**c**) Topic 3

**Figure 8.** Topic model of LNGCs aged more than 20 years. (**a**) Topic 1: tank-check-system; (**b**) Topic 2: valve-gas-renewed; (**c**) Topic 3: cargo-pump-test.

Initially, the two different tank types were separated analyzed. Since there were no significant differences in terms of key maintenance works and association with relevant maintenance works, the results were combined in single data analysis. For example, both tank types utilize submersible cargo pumps in the cargo tanks, and there was no difference in terms of maintenance-related cargo pumps or support of the cargo pumps. Rather, cargo tank leak inspection or testing received more attention, since MOSS has tank walls several centimeters in thick, whereas a membrane has about a one-millimeter thickness. Unfortunately, results have not demonstrated the difference regarding cargo tank structure including gas leak testing of both tanks types. Figure 6 illustrates the developed LNG containment system maintenance topic of LNGCs aged fewer than 10 years. As illustrated in Table 2 and Figure 5a, "check", "valve", and "cargo" were the three most frequent maintenance keywords in the same vessel age category. The first three LNG containment system maintenance topic models, however, selected topics with the keywords "check" "valve", and "tank" without a topic related to "cargo." As the selected maintenance keyword implies, the main maintenance purpose of LNGCs aged less than 10 years is the general LNG cargo handling system inspection and the regular inspection.

Figure 7 depicts the LNG containment system maintenance topic of LNGCs aged between 10 and 20 years. As illustrated in Table 2 and Figure 5b, "valve", "tank", and "check" were the three most frequently appearing maintenance keywords. However, the prior LNG containment system maintenance topic models selected maintenance topics "test", "tank", and "valve." In the second topic model in Figure 7a, the "valve"-related maintenance tasks are clustered with the "test" and "inspect"-related maintenance tasks. The relative priority of the LNG tank inspection task is higher than that of LNGCs of younger ages. Figure 7c shows that the cargo valves of the vessel age group need to be renewed and refit to ensure reliable LNG cargo handling. Overall, the prior maintenance purpose of LNGCs aged between 10 and 20 years focuses more on LNG tank inspection and cargo valve repair tasks than younger LNGCs.

Figure 8 demonstrates the proposed LNG containment system maintenance topic of LNGCs in the oldest age group. As illustrated in Table 2 and Figure 5c, "tank", "valve", and "cargo" were the most frequently appearing maintenance keywords. The developed LNG containment system maintenance topic models reflect the same tendency as the maintenance keyword frequencies. In Figure 8a, LNG tank integrity-related maintenance and inspection tasks get the highest importance for the aging LNGCs. This tendency implies that in aging LNGCs, it is necessary to pay careful attention to maintaining the hull and cargo tank integrity. Figure 8b emphasizes that the LNG cargo valves of aging LNGCs require more serious repair tasks than regular inspection. Overall, the prior maintenance

purpose of LNGCs aged more than 20 years required more maintenance than the younger LNGCs, including the LNG cargo tank and cargo handling instruments, such as valves, pumps, and cargo lines.

Table 3 outlines the proposed LNG containment system topic models with the topic summaries. The main maintenance theme of the youngest vessel age group is regular maintenance and inspection; the LNG cargo tank and cargo valves were more intensively maintained and repaired in the middle-aged LNGCs, and the oldest LNGCs required careful investigation of the integrity of the cargo containment systems.

**Table 3.** Top ranking keywords and maintenance topic summaries.

| Vessel Age | Topic Keywords | | | Maintenance Topic Model Summaries |
|---|---|---|---|---|
| | Topic 1 | Topic 2 | Topic 3 | |
| Less than 10 | check cargo gas compressor | valve pump inspection pressure | tank spare test renew | Topic 1—General inspection Topic 2—LNG handling equipment Topic 3—Tank integrity check |
| Between 10 and 20 | test inspection valve clean | tank pump cargo check | valve gas renewed refit | Topic 1—General inspection Topic 2—LNG handling equipment Topic 3—Cargo valve check |
| More than 15 | tank check system pipe | valve renewed gas inspection | cargo pump test open | Topic 1—LNG tank integrity inspection Topic 2—Valve and line inspection Topic 3—LNG handling equipment |

## 4. Discussion and Conclusions

The key principle of preventive maintenance is to conduct regularly performed maintenance to lessen the likelihood of failure of critical systems. It is important to perform while the systems are still working before serious failure happens. The proposed preventive maintenance topic models provide groups of associated maintenance works based on semantic association. If any maintenance works were identified, the person in charge of the maintenance is recommended to perform maintenance works of other items in the topic model to lessen the likelihood of failure of the associated systems.

Initially, the frequencies of maintenance keywords were analyzed to understand the relative significance of each maintenance task. The proposed maintenance topic models were not identically aligned to the maintenance keyword frequencies because the topic modeling algorithm takes into account the proximity and co-occurrence of keywords. On the basis of the proximity and co-occurrence of multiple keywords, the proposed maintenance topic models constructed meaningful and useful insights regarding the critical maintenance topics of LNG containment systems based on age categories. The proposed LNG containment topic models provided a relationship between the age and maintenance strategy of LNGCs. The relationships are categorized into the following three phases:

1. Initial phase less than 10 years after building a new ship

In the earliest LNGC life phase, condition-maintaining tasks are mainly conducted during dry dock repairs. Examples of the regular maintenance tasks include general gas detecting system inspection, gas compressor inspection, cargo valve gage pressure inspection, and cargo tank visual inspection. Moreover, the initial deficiencies during this initial operation phase are minor, and the deficiencies are generally caused by inappropriate hull design and inadequate operation of LNG transfers. Therefore, condition maintenance and regular inspection topics should be included in the preventive maintenance strategy.

2. Stable period between 10 and 20 years

In this phase, manufacturing deficiencies are limited; hence, the fundamental maintenance interest is now to ensure the acceptable and operational condition of LNG handling instruments. For this purpose, consumable spare parts, such as O-rings on many valves, bearings on rotating parts, and

parts of heaters, are renewed and refitted during the maintenance. Additionally, LNG tank integrity related to cargo tank, insulation material, and inner hull required more attention from the maintenance operators. The identified topics of the proposed maintenance topic models are therefore acceptable to incorporate these tasks.

3. Aging period after 20 years of operation

Various deficiencies and defects are caused by aging, including tearing, fatigue, and corrosion. The LNG containment system, including tanks, cargo lines, and valves, is affected by various stresses and heat shock over a prolonged time; hence, the entire LNG containment system is maintained compulsorily in order to minimize any failures. The application of newly adopted international rules and regulations must also be considered in the maintenance strategy. The recognized maintenance topics are expected to improve the established preventive maintenance of LNG containment systems by considering the aging effect of LNGCs. The proposed LNG containment system topic models contain the core maintenance tasks that should be adopted to maintain the reliable and safe operation of LNGCs of various ages.

The dramatic growth in the demand for LNG requires more LNGCs to be operational. Despite a relatively young shipping history compared with other types of marine carriers, LNG transportation retains high standards of maintenance, including a highly sophisticated containment system. As more aging LNGCs are being used, the maintenance strategy must also consider the aging effects to ensure an efficient preventive maintenance scheme. The purpose of this study was to develop preventive LNG cargo containment system maintenance models using LNGC dock specifications in LNGCs of various ages. The dock specifications for regular LNGC repairs were analyzed using NLP techniques in order to develop preventive maintenance models of the LNG cargo containment system. The results are expected to be used in the preventive LNG cargo containment system maintenance strategy in maritime LNG transportation to deliver hazardous cargoes more safely and with higher efficiency. Future work will investigate more LNGC maintenance parameters, such as repair costs, duration, and the time to develop a real-time response and cost-effective maintenance plan.

**Author Contributions:** Conceptualization, I.-H.Y.; methodology, I.-H.Y.; validation, S.-C.K.; investigation, I.-H.Y. and S.-C.K.; writing—original draft preparation, I.-H.Y.; writing—review and editing, S.-C.K.; visualization, I.-H.Y.; funding acquisition, I.-H.Y.

**Funding:** This research was funded by the Ministry of Education and National Research Foundation of Korea for a study on the "Leaders in Industry-university Cooperation Plus" project.

**Conflicts of Interest:** The authors declare no conflicts of interest.

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
