# Peer review of "Preventive Maintenance Topic Models for LNG Containment Systems of LNG Marine Carriers Using Dock Specifications"

_applsci, doi:10.3390/app9061202_

Round 1

Reviewer 1 Report

In this paper, a specific maintenance topic was found using Dock Specification. There are several questions to understand the author intend.

1. need to clarify  "Dock Specification"

The author needs to explain how the Dock Specifications is related to maintenance items regarding scheduled or unscheduled.  Most topic in the document will be related to scheduled maintenance item. If so, all of the author's works were only to categorized existing maintenance integrity.

2. 3.2 proposed Preventive maintenance topic

The author analyzed it with two different tank types such as Membrane and Moss. The maintenance philosophy of those would be different, therefore, the rank of LNG tank inspection in table 3 would be biased each other. For example, the pump in the Membrane tank is a submersible type. It was recommended that categorization of tank type is considered also.

3. 4. discussion

I would like to mention the differences in comparison with the existing schedule maintenance based topic. It is difficult to understand the difference between regular maintenance and preventive maintenance only with the topic keyword mentioned.

Author Response

<Review 1>

In this paper, a specific maintenance topic was found using Dock Specification. There are several questions to understand the author intend.

[author’s response] I appreciate the reviewer’s comment to improve the soundness of the manuscript. All revisions based on the reviewer’s comment was highlighted with YELLOW color on the original document.

1. need to clarify  "Dock Specification" - The author needs to explain how the Dock Specifications is related to maintenance items regarding scheduled or unscheduled.  Most topic in the document will be related to scheduled maintenance item. If so, all of the author's works were only to categorized existing maintenance integrity.

[author’s response] To develop the preventive maintenance model, initial research step was to identify the failure-prone LNG containment systems and to establish objective maintenance pattern from the raw data. Dock Specification was one of the available maintenance records (or data) we could achieve for the research. Maintenance works appeared on the dock specifications are mostly significant repairs and inspections, and the results of objective maintenance patterns using scheduled maintenance items were described in section 3.1. I believe the section 3.1 is closely related to the categorized existing maintenance integrity you mentioned. Additionally, LDA based topic modeling is related to preventive or predicted maintenance model by identifying frequently appeared maintenance works with associated other maintenance keywords (section 3.2).

2. 3.2 proposed Preventive maintenance topic - The author analyzed it with two different tank types such as Membrane and Moss. The maintenance philosophy of those would be different, therefore, the rank of LNG tank inspection in table 3 would be biased each other. For example, the pump in the Membrane tank is a submersible type. It was recommended that categorization of tank type is also considered.

[author’s response] Thank you for the keen attention of the different tank types. We added the above explanation of the discussion section for better clarification.

“Initially, the two different tank types were separated analyzed. Since there were no significant differences in terms of key maintenance works and association with relevant maintenance works, the results were combined in single data analysis. For example, both tank types utilize submersible cargo pumps in the cargo tanks, and there was no difference in term of maintenance related cargo pumps or support of the cargo pumps. Rather, to the cargo tank leak inspection or testing was gained more attention, since MOSS has several centimeter thicknesses of tank wall whereas membrane has about one-millimeter thickness of cargo tank was. Unfortunately, results have not demonstrated the difference regarding cargo tank structure including gas leak testing of both tanks types.”

3. 4. discussion - I would like to mention the differences in comparison with the existing schedule maintenance based topic. It is difficult to understand the difference between regular maintenance and preventive maintenance only with the topic keyword mentioned.

[author’s response] The comments regarding the difference between regular and preventive maintenance is crucial to delivering the key ideas of the submitted manuscript. We added an additional explanation on the original document.

“As the key principle of the preventive maintenance is to conduct regularly performed maintenance to lessen the likelihood of failure of the critical systems. It is important to perform while the systems are still working before serious failure happens. The proposed preventive maintenance topic models provide groups of associated maintenance works based on semantic association. If any of maintenance works were identified, the person in charge of the maintenance is recommended to perform maintenance works of other items in the topic model to lessen the likelihood of failure of the associated systems.”

Reviewer 2 Report

The paper is very well written and has a solid structure.

I only have some minor comments that could be considered by the authors.

The Introduction section is excellent, but I think it is necessary to include a few key details regarding maintenance of such vessels. What is the status quo? How often do existing LNGC require maintenance, how much  does it cost, and how many days is the vessel on lay-up for maintenance? It is important for readers that are not as familiar with such topics.

Also on Introduction, are these photos taken by the authors? There may be an issue with copyright?

Table 1: define MOSS

Section 2.1.

You should in brief present additional information on why using NLP is justified. Is the dataset of 48 vessels providing enough data to use a machine learning method? What are you trying to achieve with this method? This needs to be explained here, before you analyze how the method works.

Section 2.2 is really confusing. I do not understand what it offers to the paper.

Section 3.1. I really enjoyed this section. But perhaps you could add a comment on the importance of the quality of each maintenance report you are analyzing. On Figure 5 (and also 6,7,8) maybe include some information on the importance of font. It is obvious that the larger font means a more frequent keyword, but it is   quite qualitative in this format.

Sections 4 and 5 could be merged into one bigger section.

The main weakness of the paper is that you do not discuss repair costs, duration, and time (you do mention these as future research). So at the moment, the paper presents a method that analyzes reports, in order to indicate key maintenance items. My rather trivial question on this is whether this is actually important to use your ML method. Wouldnt this insight be also evident from a simple interview with shipowners or maintenance operators? Your method could be more useful with much larger dataset, so it would be nice if you could increase your data.

You should also add in your discussion the transferability of your paper.Can it be used in other ship types?

Author Response

<Review 2>

The paper is very well written and has a solid structure. I only have some minor comments that could be considered by the authors.

[author’s response] I would thank for the sincere comment to improve the soundness of the manuscript. All revisions based on the reviewer’s comment was highlighted with GREEN color.

The Introduction section is excellent, but I think it is necessary to include a few key details regarding the maintenance of such vessels. What is the status quo? How often do existing LNGC require maintenance, how much does it cost, and how many days is the vessel on lay-up for maintenance? It is important for readers that are not as familiar with such topics.

[author’s response] We added an additional explanation in the introduction.

“Currently most LNGC maintenance types are scheduled and corrective maintenance to maintain operational status of LNGCs. There is a lack of objective maintenance database for determining appropriate maintenance timing, particularly, LNG cargo containment systems. Dock Specification was one of the available maintenance data including significant maintenance works during a dry docking repair. All maritime vessels require to perform an inspection of the hull in a dry dock twice within 5 year period. The dry dock repair includes maintenance of hull, propeller, rudder, cargo containment systems, and other immersed part during operation. Additionally, the cost of each maintenance was removed due to the confidentiality of companies, so cost-benefit analysis was unable to perform. “

Also on Introduction, are these photos taken by the authors? There may be an issue with copyright?

[author’s response] The photos were taken by the first author, and there will be no issue with copyright.

Table 1: define MOSS

[author’s response] Modified “MOSS” to “Spherical IMO type B Tank (MOSS type)” for clarity

Section 2.1. You should in brief present additional information on why using NLP is justified. Is the dataset of 48 vessels providing enough data to use a machine learning method? What are you trying to achieve with this method? This needs to be explained here before you analyze how the method works.

[author’s response] We added an additional explanation on the original document.

“The dock specification for a dry dock repair is not created for further data analysis, so the maintenance data written in the dock specification are unstructured. Natural Language Processing (NLP) is a subfield of artificial intelligence that provides the ability to read and understand the unstructured text data from the dock specification.”

Section 2.2 is really confusing. I do not understand what it offers to the paper.

[author’s response] Section 2.2 was re-written for better information delivery.

Section 3.1. I really enjoyed this section. But perhaps you could add a comment on the importance of the quality of each maintenance report you are analyzing. On Figure 5 (and also 6,7,8) maybe include some information on the importance of font. It is obvious that the larger font means a more frequent keyword, but it is quite qualitative in this format.

[author’s response] Thank you for the keen attention of the data analysis. The reviewer’s comment was applied by adding more information regarding Figures.

“Word clouds are utilized as graphical representations of maintenance keyword frequency in dock specification. The larger the maintenance keyword in the word clouds the more common the word was in the dock specifications.”

Sections 4 and 5 could be merged into one bigger section.

[author’s response] Sections 4 and 5 were merged.

The main weakness of the paper is that you do not discuss repair costs, duration, and time (you do mention these as future research). So at the moment, the paper presents a method that analyzes reports, in order to indicate key maintenance items. My rather trivial question on this is whether this is actually important to use your ML method. Wouldnt this insight be also evident from a simple interview with shipowners or maintenance operators? Your method could be more useful with much larger dataset, so it would be nice if you could increase your data.

[author’s response] First, frequency analysis is simple enough without ML techniques. However, association analysis conducted during the topic modeling phase is complicated without machine learning because a way to determine semantic distance among words. Secondly, shipowners and maintenance operators do not have an objective rationale to preventively determine failure-prone LNG containment facilities. When authors met them for investigation of hypotheses, they were glad to get the outcomes of this research to set improved maintenance strategy of LNGCs.

You should also add in your discussion the transferability of your paper.Can it be used in other ship types?

[author’s response] The applicability of the research method of the proposed study was added in the discussion.

Round 2

Reviewer 1 Report

No further comment.